

# Linking torrential events in the Northern French Alps to regional and local atmospheric conditions

Juliette Blanchet[1], Alix Reverdy[1], Antoine Blanc[1,2], Jean-Dominique Creutin[1], Périne Kiennemann[1,3], and Guillaume Evin[4]

[1]Univ. Grenoble Alpes, CNRS, IRD, Grenoble INP, IGE, F-38000 Grenoble, France
[2]Now at RTM-ONF, F-38 000 Grenoble, France
[3]Now at Météo-France, F-74400 Chamonix-Mont-Blanc, France
[4]Univ. Grenoble Alpes, INRAE, CNRS, IRD, Grenoble INP, IGE, 38000 Grenoble, France

**Correspondence:** Juliette Blanchet (juliette.blanchet@univ-grenoble-alpes.fr)

**Abstract.** In this article we study the atmospheric conditions at the origin of damaging torrential events in the Northern French Alps over the long run, using a database of reported occurrence of damaging torrential flooding in the Grenoble conurbation since 1851. We consider seven atmospheric variables that describe the nature of the air masses involved and the possible triggers of precipitation. Using both 20CRv2c and ERA5 reanalyses, we try to isolate the variables associated with torrential events, by
objectively determining which of them differ particularly from the climatology at the dates of torrential events. This analysis is done conditionally on the main types of generating atmospheric circulation derived from Lamb weather classes, namely the North-West, Southeast-Southwest and Barometric Swamp classes. Furthermore, the atmospheric variables are considered over two spatial scales - a local scale (the Grenoble conurbation) and a regional scale (the French Alps), in order to study the spatial variability of the atmospheric signature. The results show that all atmospheric variables are less discriminant for torrential
events before 1950 according to 20CRv2c - this is likely more linked to 20CRv2c limitations over the remote past than a consequence of climate change. For the post-1950 period, similar atmospheric signatures are found both at local and regional scales in the North-West and Southeast-Southwest classes and for both reanalyses. In the North-West class - which is the best discriminated - humidity and particularly humidity transport (IVT) plays the greatest role. In the Southeast-Southwest class, instability potential (CAPE) is mostly at play. In the Barometric Swamp class both humidity (PWAT) and instability (CAPE)
are discriminant -and even more at the local scale-, showing more mixed situations generating torrential events in this class. In total, depending on the class, torrential events are 4 to 14 times more likely when the respective discriminant variables are extreme (typically above their 0.95-quantile).

## 1 Introduction

The Alps experienced through history numerous disastrous floods (Grazzini, 2007). Orography favors the combination of
abundant atmospheric precipitation and fast hydrologic concentration, driven by steep torrents and meandering rivers in flat glacial valleys. The urban areas situated in valleys are prone to combinations of torrential and riverine floods covering a wide range of vulnerable basin areas (Creutin et al., 2022).





Predicting hazard due to torrential phenomena requires understanding of the driving factors. Many studies showed the link between mesoscale heavy precipitation and atmospheric circulation. Lavers and Villarini (2013) linked mesoscale annual
precipitation maxima to the occurrence of atmospheric rivers in Europe (1979-2011). At smaller scale, Jacobeit et al. (2009) showed the link between heavy precipitation and circulation types in Central Europe (1850-2003), while Plaut et al. (2001) established strong links between mesoscale heavy precipitation and atmospheric circulation classes in the Alps for the period 1971-1995. In the Northern French Alps, Garavaglia et al. (2010) and Blanchet et al. (2021b) pointed out the Atlantic and the Mediterranean influences as the main classes of atmospheric circulation associated with extreme precipitation for the period
1950-2020 through respectively zonal and meridional flows. Blanchet et al. (2018), Blanchet and Creutin (2020) and Blanc et al. (2022a) showed that peculiar characteristics of flow strength and flow direction over western Europe drive extreme precipitation in the Northern French Alps (1950-2017). These studies were all conducted at regional scale or mesoscale, only for precipitation, and mainly for the second half of the twentieth century.

Other studies have directly addressed the link between atmospheric conditions and hydrological extremes. Prudhomme and
Genevier (2011) linked circulation type catalogues with flood occurrence for a wide range of catchment sizes (from a few tens of $km^2$ to several thousands) in Europe (1957-2002), in order to determine the most flood-producing circulation types across regions. Petrow et al. (2009), Bárdossy and Filiz (2005), Jacobeit et al. (2006) and Caspary (1995) applied the same kind of approach to catchment areas from 100 to more than 1,000 $km^2$ (in Western and Central Europe) and for hydrological return times of the order of one year. In particular, Petrow et al. (2009) and Caspary (1995) highlighted a joint change in
atmospheric circulations and intense hydrological responses. Froidevaux and Martius (2016) showed through a qualitative analysis of meteorological maps, the importance of water vapor transport in a flow perpendicular to the relief, for regional scale floods in Switzerland between 1987 and 2011. All these studies were interested in meso- to regional-scale watersheds. To the best of our knowledge only Turkington et al. (2014) studied torrential watersheds of the order of ten square kilometers, by developing atmospheric indicators allowing to isolate the situations generating debris flows and flash floods in the Ubaye
region (Southern French Alps). Among all these studies, only the study by Caspary (1995) goes back further than the middle of the twentieth century, to 1926, but for watersheds of the order of a thousand square kilometers. The study by Turkington et al. (2014) is restricted to the period 1979-2010, during which 64 flash flood events were recorded, corresponding to 6-month return periods. As far as we know, no study applied such an approach at torrential scale before the 1950s, and therefore had a sufficiently large sample of hydrological extremes to study long return periods.

In this article, we take benefit of a database of reported occurrence of damaging torrential flooding in the Grenoble conurbation (Northern French Alps) over the long run (Creutin et al., 2022). We study the atmospheric conditions at the origin of damaging torrential events in the conurbation from 1851 to 2019, in order to isolate the most generating atmospheric scenarios. Together with the extremeness of the studied events - the torrential events correspond to return periods of order 2-3 years at the scale of the conurbation -, a benefit of our work in comparison to Turkington et al. (2014) is to study the driving atmospheric
conditions with respect to the main types of atmospheric circulation. Finally, by comparing two reanalyses with different spatial resolution, we study whether the atmospheric signature of torrential events is either local (at the scale of the conurbation) or regional (at the scale of the French Alps).

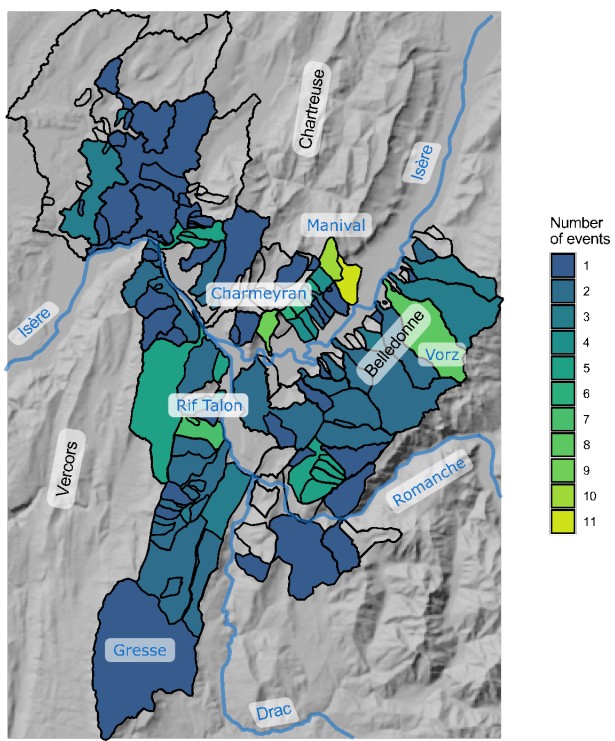

**Figure 1.** Map of the RTM torrential units of the Grenoble conurbation colored according to the number of torrential events observed over the 1851-2019 period.

## 2 Data

### 2.1 Study region

This article focuses on the Grenoble Metropolitan area, in the French Alps (see Figure 1 and its localisation in Figure 2). The torrential watersheds considered intersect the territory of the 75 municipalities included in the Grenoble Metropolitan area and/or the INSEE (National Institute of Statistics and Economic Studies) urban unit of Grenoble. This territory - called the "Y of Grenoble" because of its geometry - is located at the confluence of two rivers, the Isère and the Drac Rivers, the latter being fed further upstream by a third river, the Romanche River.

In addition to the three river sections separating the limestone mountain massifs of the Vercors and Chartreuse and the crystalline massif of Belledonne, the territory is covered by 139 torrential units as defined by the RTM (Restauration des Terrains de Montagne, a technical service of the French Forest Administration)(Figure 1). The torrential units show varied geomorphological characteristics with surfaces of less than 1 km$^2$ up to 170 km$^2$ for the Gresse torrent, despite a majority of basins of less than 20 km$^2$ (Figure 3 of Creutin et al., 2022). The altitudes are between 180 m and 2977 m.





In the Alps the sources of humidity are multiple, with much water vapor brought from the Mediterranean Sea, although there is a greater contribution from the North Atlantic in winter (Sodemann and Zubler, 2010). From a climatological point of view the "Y of Grenoble" is at the boundary of two distinct climatic zones: the Northwestern Alps and the Southwestern Alps as defined by Auer et al. (2007) using different meteorological variables. With regard to precipitation extremes, this translates into a dominance of Mediterranean flows for the massifs located to the south of our region and oceanic flows for those located

to the north of the region (Blanchet et al., 2021b). The "Y of Grenoble" is thus subject to a diversity of atmospheric influences with a strong seasonal variability (Blanchet et al., 2021a, b). This results in a wide range of atmospheric phenomena that can cause heavy precipitation locally: orographic blocking under a cold front in southerly flow or under a warm front in westerly flow, orographic lifting with dynamic northerly or westerly flows, convective developments by diurnal evolution. This justifies the need of a large sample of torrential events to characterize the generating atmospheric scenarios.

## 2.2    Torrential events

The reference for torrential events is given by the database of Creutin et al. (2022). We give below a short summary of its construction but the interested reader is invited to refer to the aforementioned reference for a deeper description. The definition of torrential events is mainly based on the database built by the RTM (a technical service of the French administration). The RTM mission of risk mapping in French mountainous areas motivated a systematic reporting for torrential and riverine

inundation by trained personal all along RTM existence. The reports provide quantitative information about the time and place of site events as well as qualitative information about the generating phenomena and the consequent damages. The reports also graduate semi-quantitatively torrential and riverine events into respectively 4 (from Very-weak to High) and 3 (from Weak to High) intensity levels. The archive of RTM reports was made publicly available during the 2010's as data sheet (https://rtm-onf.ign.fr), reporting over 30,000 site events since 1850 all over the French mountainous areas .

The selection process to define the torrential events at Grenoble Metropolitan scale simply explores chronologically the RTM database and complements it with historical research data. It looks for concurrence between site events reported by either the RTM or research data over the "Y of Grenoble" of Figure 1. A set of concurrent site events occurring during neighboring days is then defined as a Metropolitan hydrometeorological event. The isolated events of category Very-weak are discarded. The nonisolated events of Very-weak intensities are nevertheless kept - this is open to discussion, but essentially the idea is

to consider that co-occurring floods, even very weak, are "important" events. At the end, the hydrometeorological events of Creutin et al. (2022) can last from one to several days and can involve one or several rivers and/or one or several torrents. In total, the database presents 104 hydrometeorological events between 1851 and 2019.

We extract from this database 70 events involving at least one torrent (66 between 1851-2014, 42 between 1950-2014, 46 between 1950-2019). These torrential events have return periods of order 2-3 years at the scale of the conurbation but of order

6-60 years at the scale of the torrential units (Figure 1). 20% of these events took place in winter (December and February, DJF), 9% in spring (March to May, MAM), 59% in summer (June to August, JJA) and 12% in October (September to November, SON). A majority of the torrential events are multiscale in the sense that they involve multiple torrential units: only 31% of the




events involve only one unit, 21% involve two units, 9% involve three units. Finally, although they can last up to 6 days, the great majority of them (90%) last no more than 3 days (56% last one day, 24% last 2 days).

In the case of multi-day events, we associate to the event a unique "reference day" defined as the day during which the most remarkable torrential flood occurred - an expert choice (Creutin et al., 2022). This is both because some nonreference days correspond to weak events, while our goal is to link the "significant" events to their atmospheric causes, and for statistical reasons - in order to consider sequences of events of same length. In the case of 1-day events, the reference day is naturally the event day. Furthermore, to account for the fact that torrential floods can occur early or late in the day -and so be linked to

atmospheric situations the day before or after-, three-day sequences around the reference days are considered (from day-1 to day+1). We term them the "torrential events" but note that some of them differ in length from the original torrential events of Creutin et al. (2022). However it is worth noticing that actually very few days of the original torrential events of Creutin et al. (2022) are discarded with this procedure since only 10% of the original events last more than 3 days.

    Finally, each 3-day sequence between 1851 and 2019 can be flagged as either an event sequence - when centered around the

reference day of a torrential event -, or a nonevent sequence - otherwise.

## 2.3   Main types of atmospheric circulation

In order to characterize the atmospheric conditions driving torrential events, we consider two atmospheric reanalyses which are spatial and temporal interpolations of past meteorological measurements using data assimilation techniques and a meteorological model. 20CRv2c (in short 20CR, Compo et al., 2011) covers the period 1851-2014 with a spatial resolution of 2°. In this

article, we use the mean member but results with members 1 and 2 (arbitrarily tested as they are independent) are very similar (not shown). In addition, in order to study the impact of the spatial resolution, the ERA5 reanalysis (Hersbach et al., 2020), that is available only from 1950 to present, is also studied. This reanalysis has a higher resolution of 0.25° and it assimilates more data, constituting a good reference for comparison. For both databases, daily data are used.

    A classification of atmospheric circulation patterns is used to cross with the hydrological response dates of the torrential

events of Section 2.2. One of the best known and most analyzed method of classifying atmospheric circulation patterns in synoptic climatology is Lamb Weather Type classification developed for the British Isles by Lamb (1972). Its first development required subjective determination through human assessment of daily weather charts. With the advent of computer science, the classification became more objective.

    The Lamb Weather Type objective approach uses gridded daily sea-level pressure (SLP) data and was developed by Jenkin-

son and Collison (1977). The approach uses three basic variables that define the circulation features at the surface over a given window: the direction of mean flow, the strength of mean flow and the vorticity. Following previous works of Raynaud et al. (2017), Blanc et al. (2022a) and Blanc et al. (2022b) relating precipitation to surface weather and atmospheric analogues, we consider here the Western Europe window of Figure 2 and, thus, we adapt the method of Jones et al. (1993) to our region of study; we provide details in Appendix A.

The Lamb Weather Type classification contains 27 classes. Since we are here interested in the main types of atmospheric circulation, we merge the 27 classes into 5 quite balanced classes. We start from the 27 classes and we merge them according

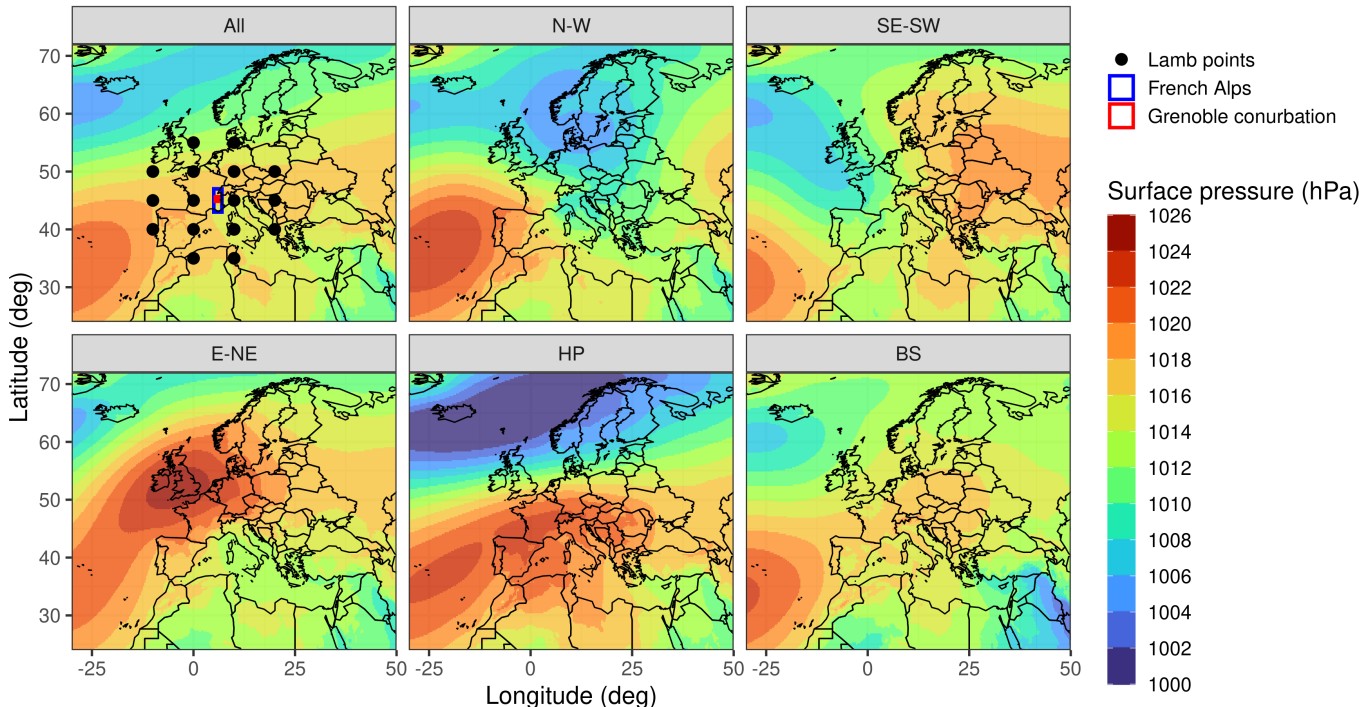

**Figure 2.** Average sea level pressure (hPa) of ERA5 over all the days of 1950-2019 (top left) and the 5 classes derived from Lamb Weather Types. In the "All" case, the black points show the 15 pixels used in Lamb classification, the blue rectangle shows the French Alps and the red one shows the Grenoble conurbation.

to their flow direction and whether they are anticyclonic or not (see Table 1). Since the no-circulation class (U) does not fit into any other class and is of sufficient size, we keep to alone. Finally, we obtain the following five classes: North to West (N-W), Southeast to Southwest (SE-SW), East to Northeast (E-NE), High pressure (HP) and Barometric Swamp (BS). We

call the "U" class "Barometric Swamp" because it corresponds to situations where there is no defined flow and the pressure is relatively uniform spatially and close to average atmospheric pressure. These are days of transition between more marked flow sequences. Figure 2 shows the average surface pressure fields in each class for ERA5 in 1950-2019. The N-W class is associated with low SLP over Scandinavia and large SLP over the near Atlantic forming a small ridge draining northwestern flows over France. The SE-SW class is associated with a trough over the northeastern Atlantic and quite high SLP over central

Europe, draining southern flows over eastern France. High SLP are centered over Great Britain in the E-NE class, while they are found between 30 and 50°N in the HP class, shifting the western flows to the north of France. The BS class is associated with a quite flat SLP field and therefore it corresponds to weak circulation over Western Europe.

The above classes are obtained at daily scale, for each reanalysis. 3-day sequences are then associated to the majority class if any (86 to 88% of the cases), or to the class of the reference (central) day otherwise. The class occurrences are given in

Table 2. Over the common period 1950-2014, the classification of 20CR mean member differs from ERA5 for 17% of the





| | |
|---|---|
| N-W: | UN, CN, UNW, CNW, UW, CW |
| SE-SW: | USW, CSW, US, CS, USE, CSE, CC |
| E-NE: | UE, CE, UNE, CNE |
| HP: | AA, AN, ANE, AE, ASE, AS, ASW, AW, ANW |
| BS: | U |

**Table 1.** The 27 Lamb classes merged into five main classes: North to West (N-W), Southeast to Southwest (SE-SW), East to Northeast (E-NE), High pressure (HP) and Barometric Swamp (BS).

| | N-W | SE-SW | E-NE | HP | BS |
|---|---|---|---|---|---|
| 20CR 1851-2014 | 13 | 25 | 18 | 36 | 8 |
| ERA5 1950-2019 | 19 | 25 | 20 | 28 | 8 |
| 20CR 1950-2014 | 12 | 24 | 18 | 37 | 8 |
| ERA5 1950-2014 | 19 | 25 | 20 | 28 | 8 |

**Table 2.** % of sequences classified in each of the 5 classes.

sequences. Differences concern mainly the N-W and HP classes that are mixed up with each other, which is not surprising since the HP class corresponds to quite flat pressure fields. However Table 3 shows that, over the common period 1950-2014, the classification of the torrential events into the N-W class is very stable - only one N-W event of ERA5 is classified into another class. It also shows that very few events are classified in the HP class after 1950 with either database and that half of the HP events of 20CR over 1950-2014 are otherwise classified in ERA5. For these reasons, the HP class is excluded from the analysis. The E-NE class is also excluded because it contains too few events (see Table 3) and it is anyway usually not associated with heavy precipitation events (Blanchet et al., 2021b). As a result, this study will focus on the N-W, SE-SW and BS classes, that contain 75 to 87% of the events depending on the reanalysis (see Figure 3). The all-events class (irrespective of the classification and including HP and E-NE events) is also considered for comparison. The seasonal distribution of the events in each class is shown in Figure 4. The great majority of events in the Barometric Swamp class occurs in summer (90% in JJA). In the other classes, events are more distributed across the seasons, with a majority of events in summer in the SE-SW class (60%) and in winter in the N-W class (50% in DJF).

## 2.4 Atmospheric variables

We consider seven atmospheric variables that describe the nature of the air masses transported and the possible triggers of precipitation: the integrated water vapor column (PWAT), the integrated vapor transport (IVT), the convective available potential energy (CAPE), the specific humidity at 700 hPa ($Q_{700}$), the horizontal wind speed at 700 hPa ($V_{700}$), the pseudo-adiabatic wet bulb potential temperature at 850 hPa ($\theta'_{850}$), and the temperature at 850 hPa ($T_{850}$). The CAPE is a proxy for atmospheric instability that can trigger precipitation (Marsh et al., 2009). Temperature could play a role in the phase of precipitation and melting. $\theta'_{850}$ summarizes the state of an air mass; it can detect anomalously warm and moist air masses (large $\theta'_{850}$) and fronts.





|  | N-W | SE-SW | E-NE | HP | BS |
|---|---|---|---|---|---|
| 20CR 1851-2014 | 24 | 15 | 5 | 12 | 10 |
| ERA5 1950-2019 | 15 | 15 | 4 | 2 | 10 |
| 20CR 1950-2014 | 13 | 14 | 4 | 4 | 7 |
| ERA5 1950-2014 | 14 | 14 | 4 | 2 | 8 |
| 20CR & ERA5 1950-2014 | 11 | 11 | 4 | 2 | 5 |

**Table 3.** Number of torrential events associated to each of the 5 classes. The "20CR & ERA5" case correspond to event that are classified in the same class with both databases.

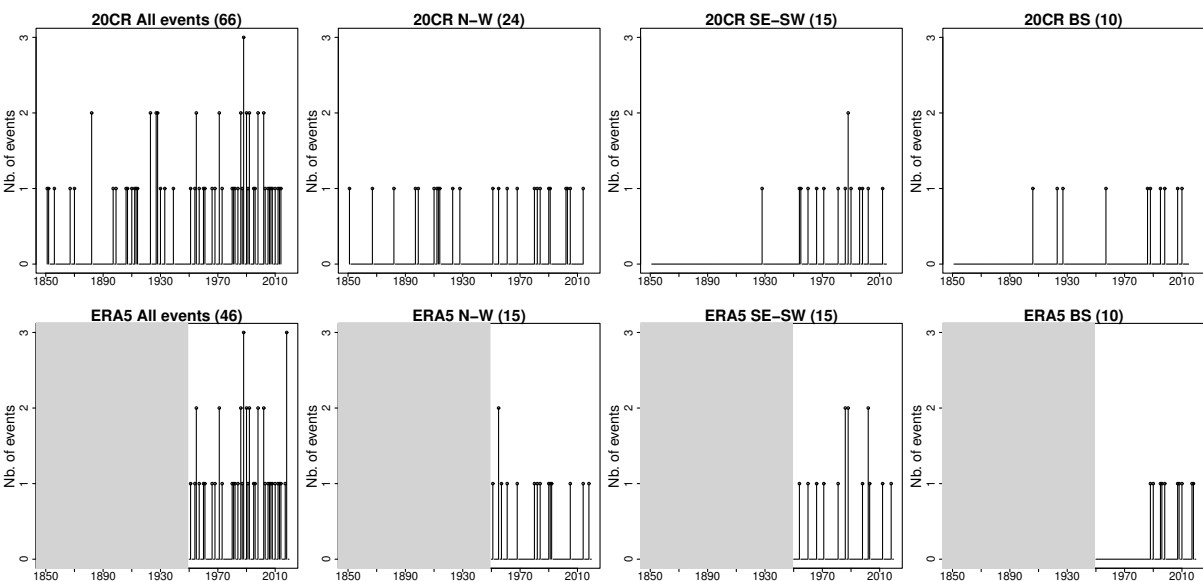

**Figure 3.** Number of torrential events per year. Top: with respect to 20CR classes over 1851-2014. Bottom: with respect to ERA5 classes over 1950-2019. The total number of torrential events is indicated in parenthesis. The "All events" case is irrespective of the classification. The gray range in ERA5 correspond to years before 1950 where ERA5 is not available.





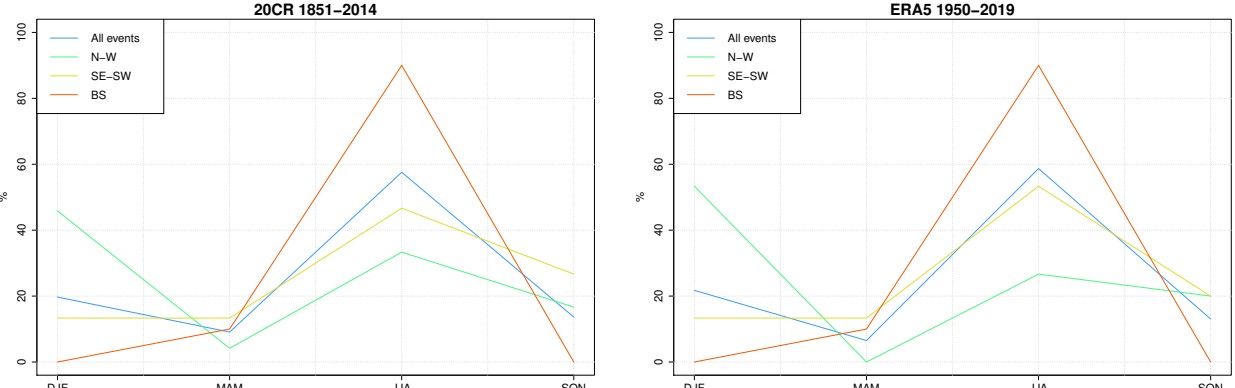

**Figure 4.** Seasonal distribution (%) of torrential events in each class. Left: for 20CR in 1851-2014. Right: for ERA5 in 1950-2019.

Tests have also been conducted on variables related to the relative humidity at 850 hPa and other atmospheric variables at 500 hPa without sufficiently conclusive results to be presented here. PWAT, CAPE, $Q_{700}$ and $T_{850}$ are directly extracted from the reanalyses, while $V_{700}$, IVT and $\theta'_{850}$ are computed following the equations in Appendix B.

In order to summarize the state of the atmosphere and simplify these two-dimensional fields (longitude, latitude), we consider daily spatial averages. Averaging is carried out on two scales in order to study the spatial variability of the atmospheric signature

of torrential events: a regional scale - the French Alps (2 pixels for 20CR, 105 pixels for ERA5), and a local scale - the Grenoble conurbation (9 pixels for ERA5). Note that, as Grenoble is at the border between 2 pixels of 20CR, we cannot consider a smaller region than the whole French Alps for 20CR.

## 3    Method

Our goal is to determine which atmospheric variables are very different from the climatology the days of the torrential events.

By "very different" we mean rare, compared to random days. Thus, instead of considering the absolute values of atmospheric variables, we consider their rarity by transforming the variables into non-exceedance probabilities (NEPs) as follows: for each atmospheric variable and for each 3-day sequence, we consider the day that experienced the maximum value. The NEP is the empirical probability of nonexceedance associated to this value with respect to all daily values. A NEP value of $90\%$ for CAPE, for example, means that the maximum CAPE of the sequence is among the $10\%$ largest daily CAPEs. In order

to account for potential seasonality, probabilities of nonexceedance are computed on either the raw daily data or the daily anomalies (substracting the daily climatology from the daily data). Comparison of the results (raw data versus anomalies) will be provided in Section 4. Considering NEPs rather than absolute values can be seen as a way of normalizing the data, since all NEPs are between $0$ and $1$.

A discriminative atmospheric variable is expected to take rare values compared to the climatology during the events and

with little variability. Following Turkington et al. (2014), we use the silhouette index, which has been widely used in clustering





techniques to assess the separation between different groups and their tightness (Rousseeuw, 1987). Let consider a given variable and a given atmospheric class, e.g. the raw CAPEs in the N-W class. We divide the corresponding NEPs into two groups: the 3-day sequences corresponding to a torrential event versus the other 3-day sequences. For a given 3-day sequence, the silhouette index determines how similar its NEP is to other NEPs in its own group compared to NEPs in the other group

(Rousseeuw, 1987). For a given 3-day sequence $s$, it is computed as

$$SI(s) = \frac{b_s - a_s}{\max(a_s, b_s)}, \tag{1}$$

where $b_s$ is the average Euclidean distance between the NEP of $s$ and all nonevent NEPs and $a_s$ is the average Euclidean distance between the NEP of $s$ and all event NEPs. Then the silhouette index of torrential events is obtained as the average silhouette index over all torrential events:

$$SI = \frac{1}{N} \sum_{s \in E} SI(s), \tag{2}$$

where $E$ is the set of torrential events and $N$ its cardinal.

Silhouette indices vary between $-1$ and $1$ but we expect them to be near-zero or positive. A silhouette value of 1 indicates that all $a_s = 0$, thus all events have exactly the same atmospheric rarity. A near-zero value indicates that all $b_s \simeq a_s$, thus the NEPs of events and nonevents are quite randomly distributed. In summary, the largest the silhouette, the more similar (grouped)

the NEPs of the events. Theoretically, the silhouette does not tell anything on the extremeness of the variable during events - which is yet our goal - since large silhouette could be obtained from nonextreme (but similar) NEPs. However, as will be seen in Section 4, all large silhouettes correspond actually to extreme NEPs so in our application the silhouette does inform on the extremeness.

The silhouette index is computed for each atmospheric class and each variable. The silhouette indices are then compared to

each other both within the class to determine which atmospheric variables are the most discriminant for the torrential events of a given class, and in-between the classes to determine which atmospheric class shows the clearest signature. However since the silhouette index is less reliable for unbalanced groups, following Turkington et al. (2014), computation follows a sub-sampling procedure: to make the event and nonevent groups balanced, we calculate the $b_s$ component in Equations 1 and 2 based on a random selection of $N$ ($=$ number of torrential events) nonevent 3-day sequences. We repeat this 1,000 times and keep the

average silhouette index over these random draws.

In order to test nonstationarity and scale effect, this procedure is applied to the atmospheric variables of four different databases:

–  20CR-1: 20CR in 1851-1949 over the French Alps (2 pixels),

–  20CR-2: 20CR in 1950-2014 over the French Alps (2 pixels),

–  ERA5-3: ERA5 in 1950-2019 over the French Alps (105 pixels)

–  ERA5-4: ERA5 in 1950-2019 over the Grenoble conurbation (9 pixels).





20CR-1 is used only for the all-events and N-W classes due to absence of events in the SE-SW and BS classes before 1950 (see Figure 3). Comparing the results of 20CR-1 to 20CR-2 allows assessing whether the atmospheric variables driving torrential events have changed over time, according to this reanalysis. We note that the two periods have different lengths (99 versus 65 years), however considering two equal periods almost does not change the results due to the absence of events in the 1930-1940s (see Figure 3). Comparison of 20CR-2 and ERA5-3 allows assessing whether different reanalyses see the same driving atmospheric variables over the recent period and the same region. Note that we have also compared over the common period 1950-2014 and the results are very similar. For shortness and to make the best use of the data, we consider here ERA5 on its full observation period. Finally comparison of ERA5-3 and ERA5-4 allows assessing whether or not the atmospheric signature of torrential events is stronger at local than regional scales.

## 4 Results

The boxplots of the NEPs during the torrential events are shown in Figure 5 for the different cases. A more quantitative comparison of the discriminative power of the different variables is provided in Figure 6 showing the silhouette indices.

Strikingly, the boxplots of all events in Figure 5 are quite widespread compared to the best variables (i.e. with the largest NEPs) of each class, which is obviously also visible in the lower silhouette indices of Figure 6. We see here the clear benefit of considering atmospheric circulation types since different variables seem to discriminate the events depending on the case. Also, differences between the raw and anomaly cases are relatively slight compared to the between-class differences. This means that the atmospheric variables driving torrential events are usually rare either overall or for the season.

For the two cases where 20CR over different periods is used (20CR-1 and 20CR-2, for the "All events" and "N-W" cases), we see a clear difference between the silhouette indices before and after 1950 (blue versus green) with a much better discrimination (larger silhouette) after 1950. However it is mainly a shift since the best variables are mostly the same over both periods. Note that the same is observed with two individual members of 20CR (not shown). We can postulate that this is more likely due to limitations of 20CR in the remote past than a consequence of climate change since there is likely no reason why *all* atmospheric variables should be more discriminant for torrential events after 1950 than before. The reduced amount of assimilated data before 1950 in the 20CR reanalysis (see Fig. 2 of Wang et al., 2012) may prevent the reanalysis from capturing the rarity of some variables, especially for $V_{700}$ and IVT for the N-W class (Figure 5), leading to lower silhouettes (Figure 6). Given these discrepancies, we were unfortunately unable to study the nonstationarity of the driving atmospheric conditions. Thus, the rest of the paper focuses on the post-1950 period (20CR-2, ERA5-3, ERA5-4).

The silhouette indices of 20CR after 1950 are relatively coherent with ERA5 for all classes but the Barometric Swamp (green versus yellow), for which 20CR finds no discriminative variable, meaning that all atmospheric variables are looser in 20CR than in ERA5 under Barometric Swamp situations, probably partly due to the rougher resolution of 20CR. For all classes, ERA5 gives similar silhouette indices over both the French Alps and the Grenoble conurbation (yellow versus red). This means that, although torrential events are triggered by very local precipitation, the atmospheric signature for such events is much wider.




**Figure 5.** Non-exceedance probability (NEP, %) for each atmospheric variable (from top to bottom: PWAT, $Q_{700}$, $T_{850}$, $\theta'_{850}$, $V_{700}$, IVT, CAPE) depending on both the used database (colors) and the atmospheric classes (panels). Top: for the NEPs of the raw data. Bottom: for the NEPs of the daily anomalies. 20CR-1 is for the French Alps over 1851-1949, 20CR-2 for the French Alps over 1950-2014, ERA5-3 for the French Alps over 1950-2019, ERA5-4 for the Grenoble conurbation over 1950-2019. There is no 20CR-1 boxplot for SE-SW and BS classes due to absence of event before 1950.


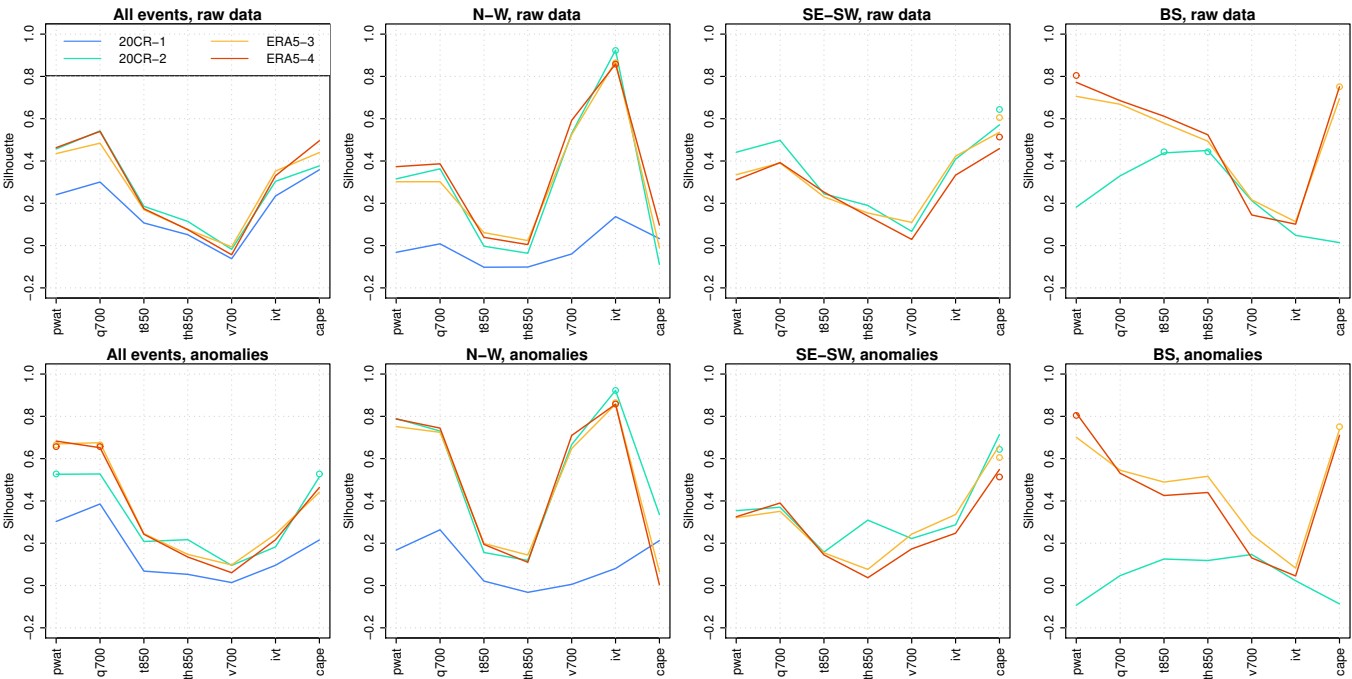

**Figure 6.** Silhouette indices for each atmospheric variable (from left to right: PWAT, $Q_{700}$, $T_{850}$, $\theta'_{850}$, $V_{700}$, IVT, CAPE) depending on both the used database (colors) and the atmospheric classes (panels). Top: for the NEPs of the raw data. Bottom: for the NEPs of the daily anomalies. The points show the largest 2d-silhouettes and the associated pairs of variables (not shown for 20CR-1).

Let now go more precisely into the different classes. Figures 5 and 6 show that, over all events, there is a quite clear signature of both humidity - large values of PWAT and $Q_{700}$, particularly for the season (anomaly) - and instability - large CAPE. Horizontal wind speed ($V_{700}$), temperature ($T_{850}$) and fronts ($\theta_{850}$) are almost not different from the climatology during the events. $Q_{700}$ and CAPE were also found to be the most discriminative variables for torrential events in the Southern French Alps in Turkington et al. (2014) (PWAT was not tested), however with lower silhouette indices (around 0.3 against 0.7 here).

One main difference is that here the silhouette is computed over the event class, whereas Turkington et al. (2014) average it over both event- and nonevent-classes. Obviously the NEP of the events in Figure 5 are better grouped (more similar) than that of the nonevents.

The influence of PWAT, $Q_{700}$, IVT and CAPE is more or less pronounced depending on the classes. In the N-W class, the humidity-related variables represented by IVT and the anomalies in PWAT and $Q_{700}$ show a very strong signature (silhouettes

above 0.8) with NEPs mainly above 90%, while the instability represented by CAPE is almost randomly distributed. Notably, the N-W class is the only one showing nonrandomly distributed - and even extreme - horizontal wind speed $V_{700}$ during the torrential events. For illustration, we show in Figure 7 the IVT fields of two events in the N-W class. Only two events are shown but this kind of pattern is found in more than the two third of the N-W events. They are associated with pronounced high pressure systems in the near Atlantic at around 35°N and pronounced low pressure systems in Central Europe. This drives


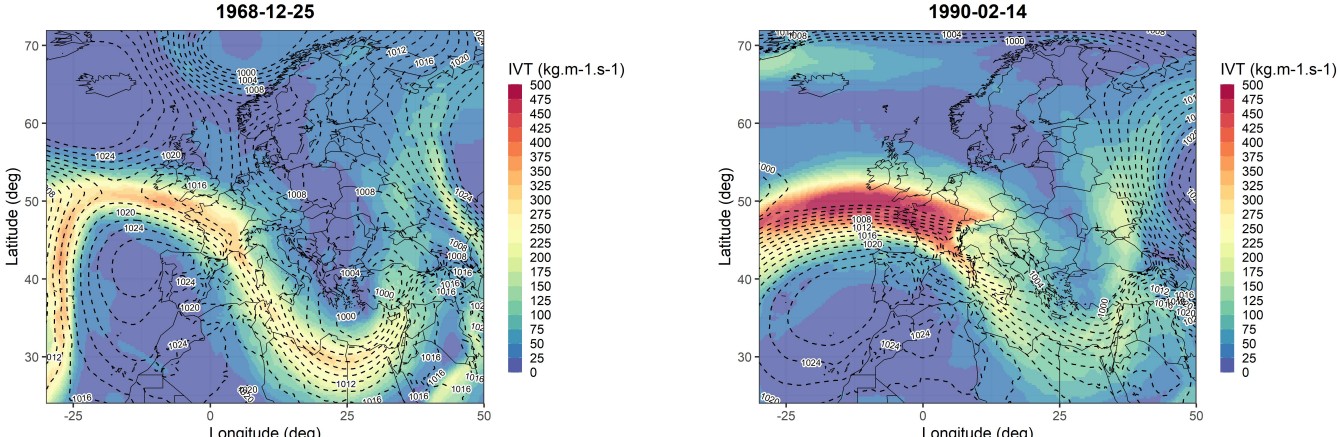

**Figure 7.** IVT fields for 2 events of the N-W class, with ERA5: 1968-12-25 and 1990-02-14. SLP are indicated on the maps and isobars are represented by dotted lines.

strong west/northwest flux associated with an elongated and intense water vapor transport towards France. Note that all events of this type in the N-W class occur in winter. The maps of Figure 7 highlight the large scale nature of the atmospheric situations driving torrential events under the N-W class in winter. This type of atmospheric situations have also been highlighted by Blanc et al. (2022a) as driving extreme precipitation over larger catchments of the French Alps (around 5000km$^2$) or by Froidevaux and Martius (2016) for floods in Switzerland, which suggests the relevance of these situations for both medium- and small-scale catchments.

The SE-SW class also shows rare values of humidity (PWAT, $Q_{700}$, IVT) but to a lesser extent (NEPs mainly above 80%), leading to lower values of silhouette. In this class, a better discrimination is provided by anomalies in CAPE, particularly at the scale of the French Alps (yellow and green), whose NEPs are mainly above 80% during the torrential events. It is difficult to find a recurrent pattern in the fields of CAPE among SE-SW events. We note that all the events associated with large CAPE values occur from late spring to late summer with a weak low pressure system located over the near Atlantic around the latitude of the Northern Spanish Coast (Figure 8, left). The other events from this class all occur from October to December with more pronounced pressure anomalies and thus larger $V_{700}$ and stronger signature of IVT, but with much smaller CAPE (Figure 8, right). This points to the variety of dynamic/thermodynamic balance through seasons in the SE-SW situations driving torrential events.

The Barometric Swamp class is somehow a mixed case where events show extreme humidity parameters PWAT and $Q_{700}$ but random IVT and very extreme instability (CAPE) according to ERA5. Notably, this is the only class where temperatures $T_{850}$ and $\theta'_{850}$ are not randomly distributed with NEPs mainly above 75%. Events from the BS class do not show recurrent spatial patterns of PWAT or $Q_{700}$, as they are by definition associated with different situations featuring a weak air circulation, as shown for two events in Figure 9. Nevertheless, it is important to note that almost all these events occur in summer (Figure 4), with therefore high air temperature allowing a large water vapor content.





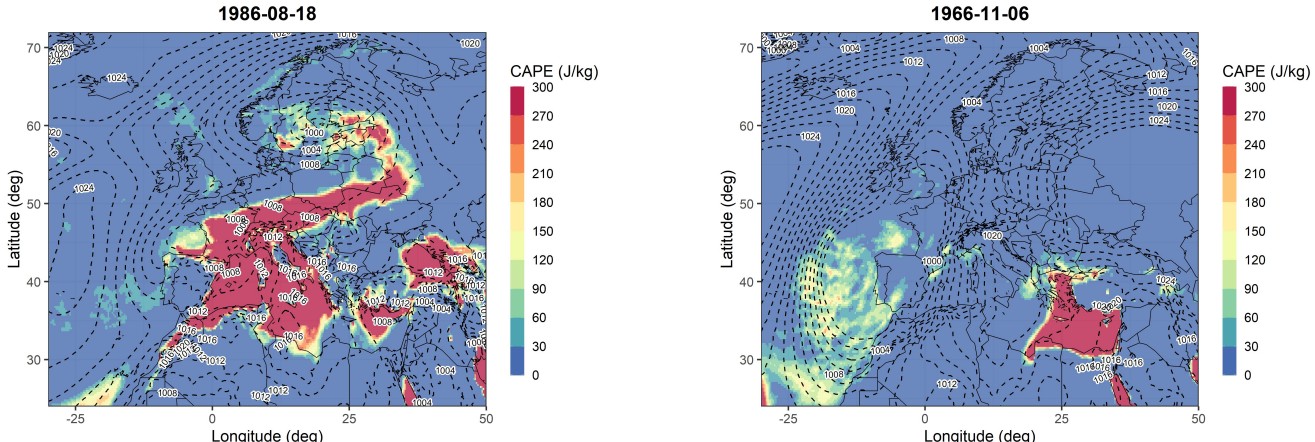

**Figure 8.** CAPE fields for 2 events of the SE-SW class, with ERA5: 1986-08-18 and 1966-11-06. SLP are indicated on the maps and isobars are represented by dotted lines.

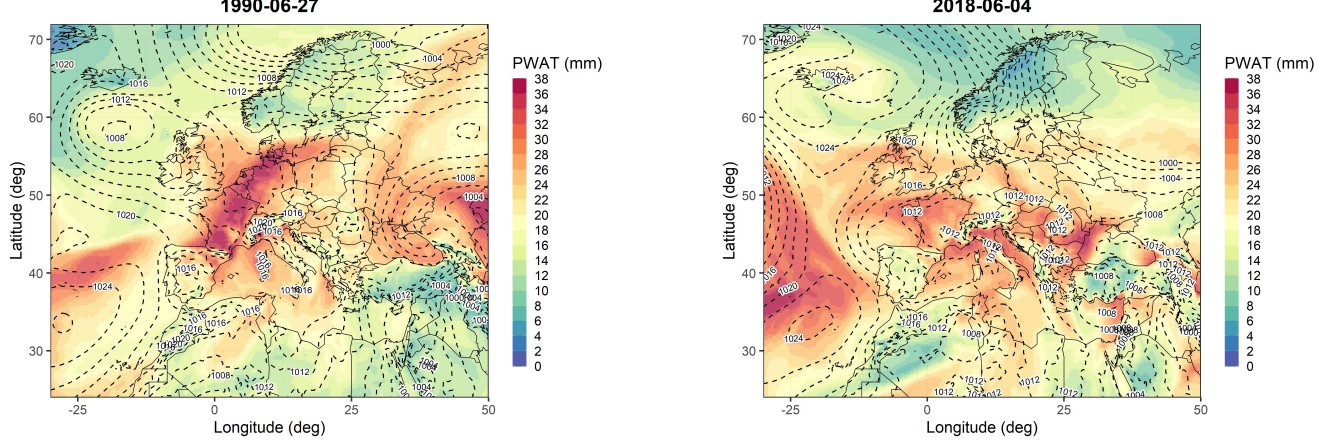

**Figure 9.** PWAT fields for 2 events of the BS class, with ERA5: 1990-06-27 and 2018-06-04. SLP are indicated on the maps and isobars are represented by dotted lines.




In conclusion the N-W events are characterized by extreme humidity and horizontal wind speed but random instability. This mainly corresponds to intense and elongated water vapor transport from the Atlantic to the Alps in winter. The SE-SW events are characterized by extreme instability but less extreme humidity, mainly occurring in late spring and summer. In comparison, torrential events occurring during Barometric Swamp situations correspond to much more mixed situations with both humidity,

instability and temperature being largely "abnormal", occurring systematically in summer. We see here the passage from large-scale atmospheric situations with a clear dynamic signature in cold seasons (N-W in winter and to a lesser extent SE-SW in autumn) to more mixed signatures between dynamic and thermodynamic (SE-SW in summer) and to a diversity of signatures in warm seasons (BS).

The same methodology can be applied to pairs of variables to study concurrent atmospheric hazards. The silhouette indices

are still obtained with Equations 1 and 2 but where the distances $b_s$ and $a_s$ are computed over pairs of NEPs. Given the seven variables of study, there are 21 different pairs of variables. Considering the raw and anomaly cases, we obtain, for every day, a total of 42 pairs of NEPs and thus 42 silhouettes. We call them the "2d-silhouettes" to remind that they are associated to pairs, but it must be noted that the values themselves are scalars. Figure 6 shows the pairs with largest 2d-silhouettes. There is no surprise here since the best pair of variables is always the two best individual variables and the associated 2d-silhouette is

almost the average of the individual silhouettes. In most cases, the best pairs correspond to a single variable associated to raw data and anomalies (e.g. raw IVT and anomaly IVT in the NE-NW class). This means that torrential events are mainly linked to a specific variable, which is either large overall (raw) or large for the season (anomaly). The three databases agree on this specific variable in the N-W (IVT) and SE-SW (PWAT) classes.

We now probabilize the occurrence of torrential events with respect to the rarity of the best pair of variables. Let $p_1$ and $p_2$ be the pair of NEPs of the two best variables. We compute the empirical probability to experience a torrential event, for a given 3-day sequence featuring NEP $p_1$ larger than a given threshold $u_1$ for the first variable and NEP $p_2$ larger than a given threshold $u_2$ for the second variable:

$$Pr(event|p_1 > u_1, p_2 > u_2).$$

The thresholds $u_1$ and $u_2$ are set to vary from 50% to 99% by step 1%. We compare these empirical conditional probabilities to the empirical probability of torrential events in the climatology by computing the ratio, in %, between the two:

$$r(u_1, u_2) = Pr(event|p_1 > u_1, p_2 > u_2)/Pr(event).$$

The corresponding ratios are shown in Figure 10 for the three databases and each class. The images show strong variations in

the top and right borders due to both the low number of events and their over-recurrence for extreme NEPs. The rows/columns showing constant values of ratios correspond to ranges that do not contain any torrential event but the rightest/topest pixel. Notably, quite similar ratios are found across databases in the N-W class, whereas the SE-SW class finds larger ratios over the French Alps (20CR-2 and ERA5-3) and the BS class over the Grenoble conurbation (ERA5-4). The largest ratios are obtained for the N-W class, as expected from the very large silhouettes in Figure 6. Torrential events are 12 to 14 times more likely

when IVT is extreme (above its 0.98-quantile) - whether at regional or local scale and whether in absolute value or anomaly. In the SE-SW class, torrential events are 7 to 8 times more likely when anomaly in CAPE is extreme over the French Alps. This





**Figure 10.** Ratios $r$ between conditional probabilities of torrential events and climatological probabilities, with respect to the considered thresholds $u_1$ and $u_2$ for the two best atmospheric variables (in abscissa is the best of the two variables according to the univariate silhouettes). Top: 20CR-2. Middle: ERA5-3. Bottom: ERA5-4. The black points show the torrential events. Note that the color scale changes from one class to another.

ratio goes down to 5 over the Grenoble conurbation. The BS class shows more variability across databases due to the smaller number of events and the discrepancies between 20CR and ERA5 (Figure 6). Torrential events are about 4 times more likely when PWAT is extreme over the Grenoble conurbation - whether in absolute value or in anomaly.



## 5  Conclusions

In this article we have studied the atmospheric conditions at the origin of damaging torrential events in the Grenoble conurbation since 1851. Using two reanalyses of different lengths and different spatial resolutions (20CR and ERA5), our study revealed i) less discriminant atmospheric conditions before 1950 according to 20CR, but which may be due to 20CR limitations in the remote past; ii) good coherence between the two reanalyses after 1950 with the exception of Barometric Swamp situations; iii) similar rarity of the atmospheric variables at local and regional (alpine) scale. Using a classification derived from Lamb weather types, we were able to show that various atmospheric conditions are favorable to torrential event occurrence depending on the type of atmospheric circulation. In the N-W class, humidity seems to play the greatest role and in particular IVT - torrential events are more than 12 times more likely when IVT is extreme. In the SE-SW class, slightly less discriminant atmospheric conditions were found with instability being mostly at play - torrential events are about 7 times more likely when anomaly in CAPE is extreme. Finally for the BS class, stronger discrepancies are found between reanalyses. ERA5 delivers the strongest signature, particularly at local scale, with BS events being characterized by mixed situations where both humidity, instability and temperature are large to extreme.

A continuation of this work will be to apply this methodology to climate projections in order to study whether the torrential flood-favorable atmospheric conditions are likely to be more frequent in the future. The fact that we were able to find discriminant signatures at regional scale is a good hope that global climate models such as CMIP6 (Eyring et al., 2016) are enough resolved to be used - whether they accurately represent the variables at play is a different story that will obviously have to be checked.

## Appendix A:  Lamb classification

The objective Lamb classification is obtained from thresholds on direction and intensity of the flow and air masses, along their vorticity. This is done using sea level pressure on 15 points of the reanalysis grid (see 2 according to the equations from Jones et al. (1993). These equations were adapted to account for the different latitudes of our study region in the latitude dependent terms, giving :





$$F_w = 0.25(P_{12} + P_{13}) - 0.25(P_4 + P_5)$$

$$F_s = 0.25a \times (P_5 + 2P_9 + P_{13} - P_4 - 2P_8 - P_{12})$$

$$F = \sqrt{F_w^2 + F_s^2}$$

$$D = \begin{cases} \tan^{-1}\left(\frac{F_w}{F_s}\right) \text{ if } F_w < 0 \\ \tan^{-1}\left(\frac{F_w}{F_s}\right) + 180 \text{ if } F_w \geq 0 \end{cases}$$

$$Z_W = 0.5b \times (P_{15} + P_{16} - P_8 - P_9) - 0.5c \times (P_8 + P_9 - P_1 - P_2)$$

$$Z_S = 0.25d \times (P_6 + 2P_{10} + P_{14} - P_5 - 2P_9 - P_{13} - P_4 - 2P_8 - P_{12} + P_3 + 2P_7 + P_{11})$$

$$a = \frac{1}{\cos\phi}$$

$$b = \frac{\sin\phi}{\sin(\phi - 5)}$$

$$c = \frac{\sin\phi}{\sin(\phi + 5)}$$

$$d = \frac{1}{2(\cos\phi)^2}$$

$$Z_{tot} = Z_S + Z_W$$

With $P_n$ the surface pressure at point $n$ (in hPa and numbered from West to East and then from North to South), $F_w$ the
355 strength of the Westerly wind component, $F_s$ the strength of its Southerly component and $F$ the total strength of the wind.
$D$ is the direction in degrees of the wind, and $Z_s$ and the $Z_w$ are respectively the Southerly and Westerly components of the
shear vorticity, $Z_{tot}$ being the total shear vorticity. $\phi$ is the latitude of the area of interest (45°N in our case for the Grenoble
conurbation). Shear vorticity and wind strength are expressed in geostrophic units (hPa per 10° of latitude at 45°N of latitude).

The wind rose is partitioned in 8 different sectors of 45° each, the first being centered on 0°. They are named according to
360 the 8 major directions of wind (N, NE, E, SE, S, SW, W, NW). A direction is attributed according to the value of $D$.

Attribution to weather classes is done according to the rules set in Table A1. The thresholds used to define the no-circulation
class (U) are adapted to our latitudes by changing the values from 6 for $Z_{tot}$ and 6 for $F$, to 4.8 for $F$ and 4.2 for $Z_{tot}$. This is
done according to Goodess and Jones (2002) to account for the lesser strength of the wind in our study region compared to the
original classification of Jenkinson and Collison (1977) for the British Isles.

**Appendix B: Atmospheric variables**

PWAT, CAPE, $Q_{700}$ (specific humidity at 700 hPa) and $T_{850}$ (temperature at 850 hPa) are directly extracted from the reanalyses.
The horizontal wind speed at 700 hPa is computed as

$$V_{700} = \sqrt{u_{700}^2 + v_{700}^2},$$




|  |  | | D | | | | | | |
|---|---|---|---|---|---|---|---|---|---|
|  |  | N | NW | W | SW | S | SE | E | NE |
| $\lvert Z_{tot} \rvert < F$ |  | UN | UNW | UW | USW | US | USE | UE | UNE |
| $F < \lvert Z_{tot} \rvert < 2F$ and $Z_{tot} > 0$ |  | CN | CNW | CW | CSW | CS | CSE | CE | CNE |
| $F < \lvert Z_{tot} \rvert < 2F$ and $Z_{tot} < 0$ |  | AN | ANW | AW | ASW | AS | ASE | AE | ANE |
| $\lvert Z_{tot} \rvert > 2F$ and $Z_{tot} > 0$ |  | CC |  |  |  |  |  |  |  |
| $\lvert Z_{tot} \rvert > 2F$ and $Z_{tot} < 0$ |  | AA |  |  |  |  |  |  |  |
| $\lvert Z_{tot} \rvert < 4.2$ and $\lvert F \rvert < 4.8$ |  | U |  |  |  |  |  |  |  |

**Table A1.** Criteria of attribution of Lamb weather classes. C means cyclonic, A anticyclonic, N North, E East, S South, W West and U is undefined in the sense that the circulation is weak to non-existent. As a result ANW means Anticyclonic with a North-West component, CC purely cyclonic, etc.

where $u_{700}$ and $v_{700}$ are respectively the zonal and meridional wind components at 700 hPa. The IVT is then computed as

$$IVT = \frac{-1}{g} \times \int Q_P \times V_P \times dP,$$

where $g = 9.81$ m/s$^2$, $P$ is the pressure, $Q_P$ and $V_P$ are the specific humidity and the horizontal wind speed at pressure $P$. Integration is computed on three levels: 850 hPa, 700 hPa et 500 hPa.





$\theta'_{850}$ (in °C) is computed from the specific humidity ($Q_{850}$), the relative humidity ($H_{850}$) and the temperature in K ($T_{850}$) at 850 hPa as follows:

$$\theta'_{850} = \theta_e - C - \exp\left\{\frac{a_0 + a_1 \times X + a_2 \times X^2 + a_3 \times X^3 + a_4 \times X^4}{1 + b_1 \times X + b_2 \times X^2 + b_3 \times X^3 + b_4 \times X^4}\right\} \tag{B1}$$

$$X = \theta_e/C$$

$$C = 273.15$$

$$a_0 = 7.101574$$

$$a_1 = -20.68208$$

$$a_2 = 16.11182$$

$$a_3 = 2.574631$$

$$a_4 = -5.205688$$

$$b_1 = -3.552497$$

$$b_2 = 3.781782$$

$$b_3 = -0.6899655$$

$$b_4 = -0.5929340$$

$$\theta_e = T_{850} \times \left(\frac{1000}{P_d}\right)^{0.285(1-0.28Z)} \exp\left\{Z(1-0.81Z)\left(\frac{3376}{T_{lcl}} - 2.54\right)\right\} \quad \text{(in K)} \tag{B2}$$

$$Z = \frac{Q_{850}}{1 - Q_{850}}$$

$$T_{lcl} = \left\{\frac{1}{T_{850} - 55} - \frac{\ln(H_{850})}{2840}\right\}^{-1} + 55 \tag{B3}$$

$$P_d = P_{lcl} - e_{sw}(T_{lcl})$$

$$e_{sw}(T_{lcl}) = 6.11\exp\left\{19.83 - \frac{5417}{T_{lcl}}\right\} \tag{B4}$$

$$P_{lcl} = 850\left(\frac{T_{lcl}}{T_{850}}\right)^{\frac{C_{pm}}{R_m}}$$

$$R_m = (1 - Q_{850})R_a + Q_{850} \times R_v$$

$$R_a = 287.04 \, J/kg/K$$

$$R_v = 461 \, J/kg/K$$

$$C_{pm} = (1 - Q_{850})C_{pa} + Q_{850} \times C_{pv}$$

$$C_{pa} = 1006.04 \, J/kg/K$$

$$C_{pv} = 1879 \, J/kg/K$$

Equation B1 is from Davies-Jones (2008), Equations B2 and B3 are from Bolton (1980) and Equation B4 is the Auguste-Roche-Magnus formula. $\theta'_{850}$ summarizes the state of an air mass. It is large for a warm and humid air mass.


*Author contributions.* JB: Formal analysis, Funding acquisition, Supervision, Writing – original draft preparation. AR: Data curation, Investigation, Visualization, Writing – review  editing. AB: Investigation, Writing – review  editing. JDC: Conceptualization, Writing – review  editing, PK: Investigation. GE: Funding acquisition, Writing – review  editing.

*Competing interests.* The authors declare there is no competing interests.

*Acknowledgements.* This study was partly funded by the European Union via the FEDER-POIA program and thanks to French national funds via the FNADT-CIMA program. This study is also part of a collaboration between the University Grenoble Alpes and Grenoble Alpes Métropole, the metropolitan authority of the Grenoble conurbation (deliberation 12 of the Metropolitan Council of May 27, 2016).



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
