# Peer review of "Linking torrential events in the Northern French Alps to regional and local atmospheric conditions"

_Natural Hazards and Earth System Sciences, 2022_

## Author Comment (AC1)

- **RC1**: 'Comment on nhess-2022-276', Anonymous Referee #1, 16 Feb 2023

The paper nhess-2022-276 "Linking torrential events in the Northern French Alps to regional and local atmospheric conditions" shows several weakness points and in my opinion should be rejected.

=> We thank the Reviewer for his/her comprehensive review. From our perspective, most of the comments made can be easily addressed. They mostly result from misunderstandings due to a lack of explanations of the main objectives of this study.

First of all, the title is not appealing: torrential events are "naturally" linked to climatic conditions.

=> We thank you for this comment. We agree there has to be some "natural" link. However it is not obvious what this link is. In particular, it is still largely unknown which atmospheric variables are at the origin of torrential events. This is precisely the goal of this study (see the first sentence of the abstract). To make it clearer, we propose to precise the title as « Linking torrential events in the Northern French Alps to regional and local **driving** atmospheric conditions »

By reading the abstract I guess that the paper is strictly related to the study area. No general conclusions are present, nothing that could be useful in places other than the study area, and this is not in line with the requirements of an international journal. Here, the aim of the paper is not well focused. Even the quotation of climate change seems not strictly related to the analysis performed in the paper.

=> This is indeed a regional analysis. For this reason, the results are not directly generalizable : without further investigation on other regions, we cannot affirm that the discriminant variables found for the study region are also the most discriminating elsewhere in the Alps or anywhere else. We anticipate it depends on the climate, the atmospheric influences, the geographical features … However the methodology used here is generic and could be used anywhere. This is for sure something we should emphasize more. In particular, the abstract could finish with « Although the results are likely to be region-dependent, the methodology used in this article is generic and could be used elsewhere to find the most discriminating atmospheric variables – provided a list of flooding dates is available ».

The only quotation to « climate change » aims at saying that the difference in 20CR results before and after 1950 is probably related to data issues. We wanted to make clear that it is not possible to conclude from these differences that the atmospheric conditions at the origin of torrential events have changed due to climate change. To make it clearer, the sentence « this is likely more linked to 20CRv2c limitations over the remote past than a consequence of climate change » could be changed to « this is likely linked to 20CRv2c issues over the remote past ».

In the introduction the problem analyzed seems the forecast of torrential events, but at the end of the introduction it seems that the authors work in a simple descriptive way, without

finalize the work to something like a classification of severity levels of the effects according to the kind of conditions that triggered them or to something else.

=> We are sorry for the misunderstanding. Forecasting is actually out of the scope of this paper. It seems we didn't used this word but "prediction", which corresponds to the computation of probabilities of occurrence. The only reference to "prediction" in the introduction aims at motivating why it is important to know what are the driving atmospheric factors, and this is precisely the goal of this article. Note that computing probabilities of occurrence such as "Pr(event| p1>u1,p2>u2)" does actually belong to the field of "prediction". In particular Fig 10 shows the "predictive gain" of considering the discriminant atmospheric variables for predicting torrential event occurrence. This can can be useful for applications such as the development of warning systems. The "prediction part" of our work is an important added-value that was not emphasized enough in the first version of the article. We will work on that in the next version of the article.

We are not sure to know what is meant by « simple descriptive way » but let stress that our study is more a statistical analysis than an event-based analysis. Rather than considering a single event and trying to understand what happened, we consider all events and try to assess in general (i.e. statistically) what were the driving atmospheric conditions. To make it clearer, we could replace the sentence « We study the atmospheric conditions at the origin of damaging torrential events in the conurbation from 1851 to 2019, in order to isolate the most generating atmospheric scenarios. » that is a bit vague by « Using a discriminative index, we statistically assess what are the atmospheric conditions driving damaging torrential events in the conurbation from 1851 to 2019. ». The two other occurrences of « study » (« study the driving atmospheric conditions » and « study whether the atmospheric signature ») could also be replaced by « assess ».

The structure of the paper is not appropriate. In the sections DATA, the authors actually just describe us what they did, and not describe the kind of data required to perform the research. And in METHOD they describe their case study. Then the paper does not supply a clear vision of what someone needs to perform the same research in another geographical area

=> Let us clarify that three types of data are used in this study (all at daily scale):

- the dates of torrential events

- a classification of atmospheric circulation into weather types

- a bench of atmospheric variables.

These three types of data are described in Sections 2.2, 2.3 and 2.4 respectively. To make it clearer we propose to :

- rename Section 2 as « Study region and data »

- reorder Section 2 as : 2.1 Study region », 2.2 Data, 2.2.1 « Dates of torrential events », 2.2.2 « Weather type classification » , 2.2.3 « Atmospheric variables » .

Please note that Lamb weather type classification is very common, this is why we consider it as « data » rather than « results » since this is not a novely of this article.

Regarding the Method section, we are not sure to see why it looks like a description of the case study. Actually Section 3 introduces the discriminative (silhouette) index and how we use it to find the most discriminative atmospheric variables. Lines 179 to 230 are actually generic and could be used anywhere else replacing « French Alps » in lines 218-219 and « Grenoble conurbation » by the respective regions. But we understand your point of view and we propose to emphasize better in the next version of the article the generality of the method and its possible use on other study regions.

Moreover, there are no sections describing in a simple way the aim of the paper and the approach followed to reach it. Maybe a flow chart of DATA and STEPS of the METHODOLOGY could help to understand.

=> This is a good idea and yes, we could provide a figure describing the different steps to make it easier to follow.

Some of the data used are actually NOT described. "20CRv2c (in short 20CR, Compo et al., 2011) covers the period 1851-2014 with a spatial resolution of 2∘. In this article, we use the mean member but results with members 1 and 2 (arbitrarily tested as they are independent) are very similar (not shown). In addition, in order to study the impact of the spatial resolution, the ERA5 reanalysis". This is simple and clear for you but not for readers that never used this kind of data and could be interested in doing. If I would like to apply the same "approach" in another study area I don't know what data I need and were could I find them!

=> 20CRv2c and ERA5 are actually common data so we had in mind that a brief description such as the previous sentence was enough: « two atmospheric reanalyses which are spatial and temporal interpolations of past meteorological measurements using data assimilation techniques and a meteorological model » (see lines 117-119). However regarding 20CRv2c members, we agree it was not clear and we propose to add after the first cited sentence : « 20CR is composed of 56 individual members that are equiprobable as well as a mean member ».

Even the information that are obtained from the analysis are sparse all around the sections and it is difficult to identify what can be useful in the practical management of torrential events.

=> Maybe the misunderstanding is that in the Data section we also give some "results":

- we give numbers about the seasonality of the torrential events (lines 99 to 104).

- we show maps of the average sea level pressure (lines 142 to 147 and Fig 2) of each class and we give numbers about the repartition of the events within the classes (lines 149 to 162 and Tables 2 and 3, Fig 3).

Regarding the first point, please note that this is more a description of the data than results *per se*, given that study of the torrential events has already been published in Creutin et al 2022 (there is nothing new here).

Regarding the second point, we agree that these are borderline between « data » and « results ». To make it clearer, we propose to move lines 142 to 162 and the cited tables/figures into a new subsection of the Result section. Section 4 would then contain 2 subsections : 4.1 « Weather types generating torrential events » and 4.2 « The atmospheric conditions driving torrential events ». In Section 4.1 we could also add a Table showing the probability of experiencing an event for a sequence in a given weather type, i.e. Pr(event) of Line 308. Thus Fig 10 would show the added discriminative power of the atmospheric variables compared to the mere classification into weather types.

Maybe something like a table of the main findings could be useful.

=> Note that the main results are given:

- in Fig 6 showing which atmospheric variables are the most discriminating for the identification of torrential flooding events and

- in Fig 10 showing how more likely it is to experience torrential flooding when the discriminant atmospheric variables are extreme.

These results are described in the text but unfortunately we don't think they can be easily described in a simple table. A summary of the discriminative atmospheric variables is provided lines 291 to 298 (« In conclusion ... »). A second summary of the results of Fig 10 would definitively be useful (after line 319). It could rephrase the last sentence of the conclusion : « In total, depending on the class, torrential events are 4 to 14 times more likely when the respective discriminant variables are extreme ».

I think that the paper must be completely re-written, putting light more on the scientific question that the paper aims to face and less on the study area (that currently is the focus of the paper). As is the paper does not show interest for an international audience.

Conclusions lack to finalize the results of the research, because not give to the reader the explanation of how these results will help in the management of torrential events or in some other framework.

=> We think there are three main contributions to our work:

- we propose a generic method to distinguish the discriminant atmospheric variables. This method could be used in other regions.

- we are able to distinguish what are the driving factors for torrential events in the Northern French Alps

- we assess what is the predictive gain provided by the discriminant atmospheric variables. This can be useful for application such as the development of warning system.

We think these points are of interest for the scientific community and suitable for NHESS. However we realize from the two Reviewers' comments that we did not emphasize these

contributions enough in the first version of the paper. We will in particular partly rewrite the abstract and the conclusion to better highlight our contribution.

Further elements in the following:

1. Figure 1 could represent something everywhere. I imagine that the authors know very well their study area but why the reader should? Geographical maps in scientific journal must have a national map inside, depicting the country where it is located, and almost the north arrow…Moreover, in the caption, RTM are quoted, but the definition of this acronym is in the next page so the reader doesn't know, at this page, what is the meaning.
   => Thanks, we will fix these points (please note that the location of the catchment within France is also shown in Fig 2, as already stated line 60)

2. Tab 1 and 2: there are no headings on the columns
   => Actually Table 1 is kind of a list : it lists the Lamb classes of each class.  Table 2 seems to show heading of the columns (the classes).

3. Table A1 should be placed after Appendix A
   => This is actually automatically placed with Latex so we let the editing to the journal.

4. The majority of formulas are not numbered
   => That's true. Our strategy was to number only the equations that are referenced to in the text but this can be easily fixed if needed.

5. It is unclear how the torrential events are identified and selected
   => This is explained lines 82 to 97 but a figure showing the different steps could be helpful.

6. L 216: actually, these are not 4 DB, but instead 2 DB split in two according to the period

   => ok, this will be modified.

7. L 223: 20CR-1 e 20CR-2 have different length but can be compared because in the period 1930-1940 no events were recorded. Could they be comparable even if some events had occurred?

   => Actually in line 223 we note that using two equal lengths (i.e. extending 20CR-2 back to the 1920s) does not change the results because it adds very few events (see Fig 3), so at the end the same discriminative atmospheric variables are found (we could show the results if needed). But please note that any period is comparable since the same methodology can be used – the question whether all periods give the same results (i.e. the same discriminative variables) is another question.

---

## Author Comment (AC2)

- **RC2**: 'Comment on nhess-2022-276', Anonymous Referee #2, 06 Mar 2023

I reviewed the paper "Linking torrential events in the Northern French Alps to regional and local atmospheric conditions".

The paper shows a structure difficult to follow, strictly related to the study area and without an international general setting.

ABSTRACT. The abstract describes what authors did, but does not report why. And above all, it does not say what can be the possible use of the results of this analysis.

=> Thank you for this comment. We agree the beginning of the abstract was too sharp. We could add a few introductory sentences saying that torrential phenomena are one of the major hazard in mountainous regions and that the driving factors of torrential phenomena are still unknown. This is precisely the goal of this article: understanding statistically what are the driving atmospheric factors.

Regarding the usability of the results, that's true this article does not have operational objectives since its goal is more knowledge oriented - understanding the driving atmospheric factors for torrential flooding. However we believe that the "predictive skill" of the discriminant atmospheric variables that is shown in Fig 10 can be useful for applications such as the development of warning systems. Flood warning is usually issued when rainfall exceeds a given threshold. However for torrent flooding we are limited because 1) the weather forecasts cannot predict precisely the location of storms, 2) we do not know enough about the hydrological functioning of watersheds to have an efficient hydrological model for forecasting. The work we propose allows us to directly attack the identification of the atmospheric factors with the most important predictive power for events that have led to damage. However we fully agree the "prediction" part of our results and its possible use was not explicitly exposed and we propose to improve this much in the next version.

Finally, please note that NHESS does not seem to publish only applications. In particular, our article is pretty much of the same vein as Turkington et al. 2014 that was also published in NHESS.

INTRODUCTION. In the introduction, the papers quoted as "state of the art" are mainly old or very old (the only recent references concern the effects of precipitations (Creutin 2022) and not the precipitations and atmospheric circulation).

=> Actually it seems we also cite recent papers such as Blanchet et al 2021b, Blanchet et al. 2018, Blanchet and Creutin 2020, Blanc et al 2022a. However we agree all these are from coauthors of this study and very recent international references are missing. The other articles we cite are more from the 2000s-2010s (Jacobeit et al 2009, Plaut et al. 2001, Garavaglia et al 2010), which doesn't seem to us to be out of date but we could also cite e.g. Lenggenhager & Martius, 2019, Giannakaki & Martius, 2016 and Mastrantonas et al., 2021.

At the end of this section, describing the aim of the paper:

*"Together with the extremeness of the studied events - the torrential events correspond to return periods of order 2-3 years at the scale of the conurbation -, a benefit of our work in comparison to Turkington et al. (2014) is to study the driving atmospheric conditions with respect to the main types of atmospheric circulation"*. It is a little unclear as aim of the paper.

=> We mean here that depending on the type of atmospheric circulation, different atmospheric variables can be at play and this is what we will study here. In Turkington et al 2014 all events were analyzed together, irrespective of the type of circulation. In other words, the study of Turkington et al 2014 is comparable to the "All events" case in Figs 5, 6 and 10. So considering the N-W, SE-SW and BS circulation types is a novelty of our approach. However, we agree with the reviewer that this sentence is unclear and it will be rephrased.

Figure 1 does not present reference elements, it is obscure where this area is.

=> Please note that the location of the area is shown in Figure 2, as written line 60. However to make it clearer, we will add a map of France in Figure 1.

DATA. Why a reader should search a figure published in another paper as suggested here? *"The torrential units show varied geomorphological characteristics with surfaces of less than 1 km2 up to 170 km2 for the Gresse torrent, despite a majority of basins of less than 20 km2 (FIGURE 3 OF CREUTIN ET AL., 2022)"*.

=> We don't think it is useful to represent Figure 3 of Creutin et al 2022 in this article since the figure merely shows the distribution of surfaces, which is already visible qualitatively in Figure 1. Thus to avoid confusion, we could omit the full reference (removing the parenthesis above) and refer to Figure 1 of the article.

TORRENTIAL EVENTS. Here the description is verbose: a table could have been more explicative.

=> We guess the Reviewer refers to lines 98 to 104? If so yes, this is a good idea, we could add a table.

Here: *"…the hydrometeorological events of Creutin et al. (2022) can last from one to several days and can involve ONE OR SEVERAL RIVERS AND/OR ONE OR SEVERAL TORRENTS. In total, the database presents 104 hydrometeorological events between 1851 and 2019. We extract from this database 70 events INVOLVING AT LEAST ONE TORRENT"*. In my opinion this is contradictory or not properly explained…

=> Let's put it another way. The events can either involve: a) one or several rivers and no torrent (purely riverine events), b) one or several torrents and no river (purely torrential events), c) one or several torrent and one or several rivers (mixed events). In total there are 104 events among which 70 belong to b) and c) i.e. involve at least one torrent. We will rephrase the sentence to make it clearer.

Moreover: *". These torrential events have return periods of order 2-3 years at the scale of the conurbation but of order 6-60 years at the scale of the torrential units (Figure 1)"*. This should be explained, and moreover in figure 1 there are no mention of return periods.

=> Figure 1 shows the number of events per torrential unit, so dividing each number by 169 (numbers of years in 1851-2019) gives an approximation of the return periods. So 11 events correspond to about one event in 15 years and 1 event correspond to about one event in 170 year. At the scale of the conurbation, 70 events corresponds to about 1 event in 2-3 years. We will correct the numbers (6-60 years → 15-170 years) and explain them better.

"*Finally, each 3-day sequence between 1851 and 2019 can be flagged as either an event sequence - when centered around the reference day of a torrential event -, or a nonevent sequence – otherwise.*" unclear…

=> An example might be easier to follow. Let consider the 3-day sequence  from 1968-12-24 to 1968-12-26. Then 2 possibilities: 1) If 1968-12-25 (the central day) is the reference day of an event, then the 3-day sequence is an "event sequence". If 1968-12-25 is not the reference day of an event, then the sequence is a "nonevent sequence". We will add a similar example in the article.

MAIN TYPES OF ATMOSPHERIC CIRCULATION: "…*we consider two atmospheric reanalyses which are spatial and temporal interpolations of past meteorological measurements using data assimilation techniques and a meteorological model. 20CRv2c (in short 20CR, Compo et al., 2011) covers the period 1851-2014 with a spatial resolution of 2∘. In this article, we use the mean member but results with members 1 and 2 (arbitrarily tested as they are independent) are very similar (not shown)*".

This is the only (not sufficient) explanation about data used, that are "described" in a section concerning atmospheric circulation, not DATA USED. So in this section there is a confusion between DATA, DESCRIPTION OF STUDY AREA CHARACTERISTICS and RESULTS OF THE ANALYSIS, with also literature quotations concerning approaches to study atmospheric circulation… it is very complex and not in line with the classical way in which scientific paper are structured (dividing data from method and results).

=> Actually we consider the atmospheric circulation type as "data" because Lamb weather type are very common, so this is not a novelty of our paper. Thus three types of data are used in this study (all at daily scale):

- the dates of torrential events

- a classification of atmospheric circulation into weather types

- a bench of atmospheric variables.

These three types of data are described in Sections 2.2, 2.3 and 2.4 respectively. As also answered to Reviewer 1, to make it clearer we propose to :

- rename Section 2 as « Study region and data »

- reorder Section 2 as : 2.1 Study region », 2.2 Data, 2.2.1 « Dates of torrential events », 2.2.2 « Weather type classification » , 2.2.3 « Atmospheric variables » .

We also propose to add [lines 142 to 147 and Fig 2] and  [lines 149 to 162 and Tables 2 and 3, Fig 3] into a new subsection of the Result section to better distinguish between data and

results. The Results Section would contain 2 subsections : 4.1 « Weather types generating torrential events » and 4.2 « The atmospheric conditions driving torrential events ».

METHODOLOGY. The methodology is not clearly described as well as the data sources and data types used. The very large number of acronyms not reported in a list make very difficult to follow the paper.

=> The methodology is described in Section 3 and, without precision, we are not sure to know what is "not clearly described". The data sources and data types are presented in the preceding section, Section 2 "Data".  However we agree a table with acronyms could be useful and we will add one.

I understand that the authors are treating a large amount of data and it is difficult to summarize results of the analysis but they lack either clarity or consistency in presentation.

The aim of the paper, the research question/s and the possible uses (especially outside the study area) are not clearly stated.

=> This is indeed a regional analysis. For this reason, the results are not directly generalizable : without further investigation on other regions, we cannot affirm that the discriminant variables found for the study region are also the most discriminating elsewhere in the Alps or anywhere else. We anticipate it depends on the climate, the atmospheric influences, the geographical features … However the methodology used here is generic and could be used anywhere. This is for sure something we should emphasize more. In particular, the abstract could finish with « Although the results are likely to be region-dependent, the methodology used in this article is generic and could be used elsewhere to find the most discriminating atmospheric variables – provided a list of flooding dates is available ». We could also add a similar sentence in the conclusion.

As explained above, the goal of the paper is to find the atmospheric factors driving torrential events in the Northern French Alps. We aim physical understanding more than a direct application. However as already said above, the results of Fig 10 showing the gain in "predictive skill" provided by the discriminant atmospheric variables can be useful for applications such as warning. However we agree the "prediction" part of our results was not clearly exposed and we propose to improve this much in the next version.

The method starts with: "*OUR GOAL IS TO DETERMINE WHICH ATMOSPHERIC VARIABLES ARE VERY DIFFERENT FROM THE CLIMATOLOGY THE DAYS OF THE TORRENTIAL EVENTS.*" Maybe there is something missing because I cannot understand this aim. Climatology for me is a discipline but maybe I'm wrong.

=> In the literature, "climatology" is sometimes used to refer to the average conditions of the atmosphere (e.g. the mean daily precipitation of each calendar day) . Here, it means that we determine which variables are different the days of the events compared to "usual days" and it will be precised.

After this, the methodology is explained in a confuse manner, presented more like a description of what the authors did then like "how to do to perform the same experiment

in another area". The steps of the methodology are not described, it seems made of a series of actions having no linear path, without a chronological sequence to follow.

=> All the steps of the methodology seem to be described in Section 3 but, as proposed by Reviewer 1, the steps of the methodology will be made clearer with a figure.

In the method sections, parts of data elaboration are included and the reader is unable to understand if these questions are related to the methodology or depends on the specific data used (*"We note that the two periods have different lengths (99 versus 65 years), however considering two equal periods almost does not change the results due to the absence of events in the 1930-1940s (see Figure 3). Comparison of 20CR-2 and ERA5-3 allows assessing whether different reanalyses see the same driving atmospheric variables over the recent period and the same region. Note that we have also compared over the common period 1950-2014 and the results are very similar"*).

=> Lines 216 to 230 could be moved to the end of Section 2.4 "Atmospheric variables" since it is indeed related to the data. As already said, the methodology is generic and it is not related to the data nor the study region. We will make this clearer.

Conclusions do not explain the way in which the results obtained can be practically or theoretically applied.=> The results are probably not directly applicable, however we do think that understanding what are the driving factors is of interest for the scientific community and suitable for NHESS. Also the predictive gain provided by the discriminant atmospheric variables can be useful for applications such as the development of warning systems which could be based on the most discriminant atmospheric variables that corresponds to parameters routinely produced by the numerical weather forecast systems.. Finally, an added-value of our article is also to provide a generic method that could be used in other region to find the discriminant atmospheric variables and to compute the corresponding predictive skills. However we agree we did not highlight these points enough and we will emphasize them much more in the next version.

In my opinion this paper cannot be accepted.

From our perspective, most of the comments made by the reviewer can be easily addressed. They mostly result from misunderstandings due to a lack of explanations of the main objectives of this study.

References

Giannakaki, P. & Martius, O.
Synoptic-scale flow structures associated with extreme precipitation events in northern Switzerland
*International Journal of Climatology*, **2016**, *36*, 2497-2515

Lenggenhager, S. & Martius, O.
Atmospheric blocks modulate the odds of heavy precipitation events in Europe
*Climate Dynamics*, **2019**, *53*, 4155-4171

Mastrantonas, N.; Herrera-Lormendez, P.; Magnusson, L.; Pappenberger, F. & Matschullat, J.
Extreme precipitation events in the Mediterranean: Spatiotemporal characteristics and connection to large-scale atmospheric flow patterns
*International Journal of Climatology,* **2021***, 41,* 2710-2728

Turkington, T.; Ettema, J.; van Westen, C. J. & Breinl, K.
Empirical atmospheric thresholds for debris flows and flash floods in the southern French Alps
*Natural Hazards and Earth System Sciences,* **2014***, 14,* 1517-1530